# Female zebrafish (*Danio rerio*) demonstrate stronger preference for established shoals over newly-formed shoals in the three-tank open-swim preference test

**Andrew J. Velkey** [1]*, **Caroline H. Koon**[1], **Isabel A. Danstrom**[1], **Katie M. Wiens**[2]

1 Neuroscience Program, Christopher Newport University, Newport News, VA, United States of America,
2 Science Department, Bay Path University, Longmeadow, MA, United States of America

☯ These authors contributed equally to this work.
* avelkey@cnu.edu

## Abstract

Zebrafish (*Danio rerio*) share a considerable amount of biological similarity with mammals, including identical or homologous gene expression pathways, neurotransmitters, hormones, and cellular receptors. Zebrafish also display complex social behaviors like shoaling and schooling, making them an attractive model for investigating normal social behavior as well as exploring impaired social function conditions such as autism spectrum disorders. Newly-formed and established shoals exhibit distinct behavior patterns and inter-member interactions that can convey the group's social stability. We used a three-chamber open-swim preference test to determine whether individual zebrafish show a preference for an established shoal over a newly-formed shoal. Results indicated that both sexes maintained greater proximity to arena zones nearest to the established shoal stimulus. In addition, we report the novel application of Shannon entropy to discover sex differences in systematicity of responses not revealed by unit-based measurements; male subjects spent more time investigating between the two shoals than female subjects. This novel technique using established versus newly-formed shoals can be used in future studies testing transgenics and pharmacological treatments that mimic autism spectrum disorder and other disorders that affect social interaction.

## Introduction

Zebrafish share many relevant genes with mammals, promoting the species as a useful model for neuroscience, biomedical and human behavioral disorder research [1, 2]. Zebrafish have been used as an animal model for disorders of the nervous system including anxiety [1, 3], addiction [4], epilepsy [5], and several neurodegenerative diseases [6, 7]. Zebrafish also demonstrate sociability and many aspects of grouping behaviors [8–10] and are therefore a viable model for investigating social behavior as well.

Zebrafish display a strong preference towards joining a shoal with live conspecifics versus remaining socially isolated [11–14]. Group living enhances survival through multiple anti-

**Funding:** We are grateful for the Christopher Newport University Office for Undergraduate Research and Creative Activity, which provided financial support of this work through the CNU Summer Scholars Program and the Undergraduate Research Captains Program. These programs provided housing and stipends to the second author (CHK) in support of pilot work, data collection, and manuscript preparation.

**Competing interests:** The authors have declared that no competing interests exist.

predatory defense mechanisms. Due to the many eyes hypothesis, the predation risk of shoaling can be minimized by the benefits of aggregation, as a large group is better equipped than an isolated animal to detect predation compared [15, 16]. Heightened shoaling behavior could also be an indicator of positive affect as it can be spontaneous and involves fewer antagonistic interactions [17]. However, group living can also provide adverse circumstances, as parasites are more easily spread in large groups, and large groups can be more prone to targeting by predators [18]. In addition, intra-group competition for food and resource limitation grows as shelter size increases [19]. It is necessary to determine affinity for specific shoal characteristics to completely understand the nuances of this complex behavior and apply it to human disease models.

While many studies have addressed zebrafish preference based on visual characteristics such as fish size [20], shoal size, male-to-female ratio, stripe pattern, etc. [21], less data exists regarding the role of intragroup familiarity in zebrafish social preference. Our results indicate that a single test subject can differentiate between an established shoal and a newly-formed shoal, and both male and female subjects prefer to spend more time in proximity with the established shoal. Female test fish explore more within the vertical column nearest the established shoal while male test fish make more cross-tank transitions.

## Zebrafish exhibit preferences in social choice

Previous studies indicate characteristics driving fish shoal preference such as shoal size [22], shape [23], and parasite load [24]. Visual cues specific to individual fish within shoals is also an important determinant of shoaling behavior. Zebrafish are able to identify phenotypic differences in stripe pattern and exhibit shoaling preference dependent on early exposure to specific patterns that do not depend on their own phenotype [25, 26]. European minnows prefer to interact with shoals that are known to them [27] even if the familiar shoal is the smaller shoal; however, no research has presented subjects with a choice between novel shoals that have different levels of inter-member familiarity. Social preferences based on visual cues suggest that shoal features may also influence preference of a lone fish [28].

**Sex differences for shoaling preference.**   Certain aspects of shoaling behavior differ between males and females [29]. Interestingly, male zebrafish display bolder responses than females in both the open-field test and the novel object test [30], and male zebrafish are more exploratory of novel environments than females [31]. Shoal sex and size represent two qualities that may influence the affinity a subject has for a particular shoal. Previous studies have supported the presence of sex differences in zebrafish involving choice between joining a shoal or remaining segregated [32]. In one study, a single subject chose to spend time in proximity with a singular same-sex fish than with a conspecific of the opposite sex [13, 33]. However, another study that observed subject partiality between shoals demonstrated that males preferred to shoal with females, though females demonstrated no clear preference to shoal with one sex over the other [13]. Though male zebrafish have demonstrated a lack of significant preference for a particular shoal size, females display a clear preference for larger shoals compared to smaller ones [12, 13]. Because females may seek shoaling for protection purposes, prioritizing shoal size over shoal sex is anticipated in female shoaling behavior; larger shoals provide greater protection from predators through enhancing the confusion effect [22, 32] and increasing the dilution effect [33, 34], thus affording greater protection for each member of the shoal [12]. Previous research indicates that predation risk is a likely reason for female sexual segregation [31]. The dynamics and safety of shoal stability may also impact a female's shoal preference when shoal size is modulated [12]. However, both male and female zebrafish prefer to affiliate with shoals rather than remain socially isolated, emphasizing innate social

tendencies commonly observed among the species [12, 13]. Because both females and males show preferences for different shoal characteristics such as size and collective group shape, additional factors likely influence zebrafish shoaling behavior; other criteria may include parasitism [35], distance from predators [36], and presence of poor competitors [37].

**Novel vs. established shoals.** A newly formed shoal generates immediate competition between members, resulting in the formation of dominant and subordinate roles within the shoal [38]. A previous study examined the activity of both dominant and subordinate male zebrafish immediately after grouping and again after five days of acclimation to conspecifics. Behavioral observations on the fifth day suggest that the formation of a stable social hierarchy occurs within the first five days following initial group formation [39]. These visible alterations in behavioral tendencies after shoal formation suggest that an established shoal may appear as a less threatening environment, while a newly-formed shoal may present as a more harmful environment due to the lack of established hierarchical roles. In addition to known sexually dimorphic behavioral characteristics in zebrafish [40, 41], dominance and aggression behavior patterns within shoals might differ between males and females. For instance, dominant males are more aggressive with their submissive conspecifics than dominant females [42], and males demonstrate stronger lateralization during aggressive responding than females [43].

Behavioral markers of previously established shoals have been observed and characterized. The Trinidadian guppy (*Poecilia reticulata*) and fathead minnow (*Pimephales promelas*) exhibit shoaling behaviors similar to those displayed by zebrafish and are also used as models for sociality and social behavior [44, 45]. Findings from investigations on Trinidadian guppy shoal fusion indicate a gradual decrease in the mean difference in shoal member size after two shoals were introduced. Nearly all previously established shoals reformed new groups due to fish size preference [46]; the shoal fusion that took place illustrates the capacity of fish to choose a shoal based on member characteristics. Previous explorations of fathead minnows have revealed variations in behavioral responses under conditions that are indicative of a predatory threat [47]. Through observing behaviors of familiarity, or lack thereof, one can determine whether a newly-formed or established shoal is more stable when presented with a potential predator. Established shoals with intergroup familiarity demonstrated more tight shoaling behavior, less freezing, and more dashing behavior. Additionally, these members performed more inspection visits compared to subjects in the newly-formed shoal [47]. These findings support the notion that a previously established shoal provides heightened security during potential threats of predation. Based on the increased likelihood of survival, it is possible that zebrafish and minnows may choose to join a shoal that presents as established. In addition, previous research suggests that intragroup familiarity is accompanied by more efficient communication between members, providing a less dangerous social living space [48, 49].

Shoal cohesion is characterized by interactions between conspecifics in response to their changing environment [50]; therefore, observing intermember communications may help determine the capacity for different shoals to attract isolated fish. Recently, investigators examined the preference of both male and female wildtype zebrafish upon providing subjects with the choice between a familiar fish and a novel fish. Both male and female subjects exhibited a preference for a novel conspecific rather than a familiar one [11]. The implications from this study strengthen the conjecture that zebrafish possess social memory. It is therefore likely that swimming pattern and tank localization can serve as dependent variables to identify affinity toward shoals of varying levels of establishment.

We suggest that, in conjunction with the ability to differentiate between shoals, zebrafish are capable of detecting the degree of polarization of a particular shoal by observing swimming patterns of the shoal members. Previous investigations have observed longitudinal patterns of shoaling and schooling of zebrafish to identify characteristic differences of the two behaviors

[51]. Polarization represents the tendency of a group to swim in the same direction. Based on differential characteristic swimming patterns, shoals are seen as a low-polarized group, while a school is considered a high-polarized group [16, 52]. When relative location and average movement velocity of subjects were analyzed over a five-day period, polarization decreased after this interval, possibly due to a higher level of perceived comfort or safety [53]. These findings raise the question of whether acclimatizing to an established shoal appears more attractive to a single test fish rather than acclimatizing to a newly formed, and possibly more polarized, shoal. Namely, will a new zebrafish "fit in" better with an established shoal because it is seeking comfort and safety?

Prior research has not only considered the innate social tendency of zebrafish to shoal [29] but has also investigated numerous aspects of zebrafish behavior within a shoal [47, 52, 54]. Nonetheless, few studies have examined the role of cohesion and shoal stability in determining preferences between shoals. Further, though some studies support the conjecture that zebrafish exhibit evidence of social memory [11, 55], it is still unknown whether a lone fish can perceive the visual differences of a newly-formed versus an established shoal, and how those differences might affect social choice. Prior research in our lab demonstrated the experimental efficacy of a novel open-swim paradigm for studying zebrafish social preference [56]. We used the open-swim task to test zebrafish preference for an established shoal over a newly-formed shoal. The test fish was placed in the center compartment of a three-chamber tank system while both an established and a newly-formed shoal were each displayed in the two flanking tanks. Given the reduced aggression and increased cohesion found in established shoals, both male and female zebrafish subjects demonstrated proximal preference for an established shoal over a newly-forming shoal. Furthermore, notable sex differences were demonstrated with female zebrafish showing stronger preference for established shoals over newly-formed shoals while also displaying lower behavioral entropy than male zebrafish. The results from the current study expand the present knowledge on shoaling preference and can be used in future studies of social preference in wild-type as well as transgenic lines of zebrafish.

## Method

### Subjects

The experimental subjects ($N$ = 82) were healthy mature male ($n$ = 45) and female ($n$ = 37) wild-type zebrafish (EKK strain) of approximately 6–12 months of age and 2.5–6.4 cm in length. Experimental subjects as well as the fish used as stimuli were obtained from Aquatica Tropicals, Inc. (Ruskin, Florida, USA). Velkey et al [56] used a total of 78 subjects (males and females with no tests for sex differences) in conditions testing a live-shoal stimulus (vs. either a video stimulus or a mobile-model stimulus) in their study which revealed significant preferences for live-shoal stimuli. As such, a similar sample size was used for the current study. All care and treatment of subjects in the present study were consistent with the recommendations in the Guide for the Care and use of Laboratory Animals [57]. The research was conducted under an existing protocol (#2019–8) reviewed and approved by the Christopher Newport University Institutional Animal Care & Use Committee.

### Materials, apparatus, and procedure

Subjects were sexed and housed in eight separate holding tanks (20.3 x 30.5 x 50.8 cm), each holding 37.8 liters (10 gal) of conditioned water maintained at a temperature of 28.5˚ C. Visual barriers were placed between holding tanks to avoid further familiarity. Four groups of males and four groups of females were housed in the eight holding tanks so that each tank consisted of an equal number of fish. The maximum fish density in the holding tanks was approximately

1.5 fish per gallon. The water had constant filtration and aeration systems, and all fish were housed under a 14-hour light/10-hour dark cycle. Fish were fed daily using the Aquaneering Scientific Hatcheries Diet for *Danio rerio*. Fifty percent water changes were performed weekly for all housing and experimental tanks. The present study used newly-formed and established shoals as stimuli. Novel shoals were grouped in the morning and used the same day of experimentation as a stimulus; to ensure the demonstration of behaviors that characterize a shoal as novel, each shoal member was randomly selected from one of the four separate holding tanks. Shoals were held for seven days in groups of four fish in order to establish intra-shoal familiarity [58], and these shoals were subsequently used as the established shoal stimulus.

The objective of the present study was to determine the behavioral responses of test subjects when they were presented with different shoaling stimuli. Experimental subjects were selected for each trial and performed an open-tank, free swim task [11, 56]. The testing tank (20.3 x 25.4x 40.6 cm) was one 20.8liter (5.5 gal.) tank positioned between two stimulus tanks of the same size (Fig 1). Rosco brand Linear Polarizing Filter Sheets (#7300) were obtained from B & H Photo and Video (New York, NY) and were placed on the outer surface of the center tank between the side tanks. One of the filters had the grid oriented horizontally and the other grid was vertically oriented. With the filters in place, the stimulus fish of the tank on one end were unable to see the stimulus fish on the opposite end, but the subject in the center tank was able to see the stimuli in each of the flanking tanks. The two stimulus shoals each contained four zebrafish, all of the same sex as the test subject, but each fish was taken from a different home tank. A novel shoal was formed using one fish from each of the four same-sex holding tanks. In seven days, that same shoal was considered established and reused in trials as an established stimulus shoal. After an established shoal was used in one trial, it was added to a tank for later use so that each of the fish could be individually utilized as a subject. For each stimulus pairing, the position of each stimulus type was counterbalanced such that an equivalent number of trials were run with each stimulus on the left side as the number of trials with each stimulus on the right side. The two stimuli in each flanking compartment consisted of a same-sex newly-formed shoal on one side and a same-sex established shoal on the opposite side.

Behavior tracking and analysis was conducted using EthoVision XT 15.0. Acquisition of tracks of the test subject via EthoVision XT 15.0 initiated after a 3-minute habituation period. Prior to and during the habituation period, the lateral sides and rearmost wall of the experimental tank were opaque in order to reduce the influence of the surrounding area. The lateral

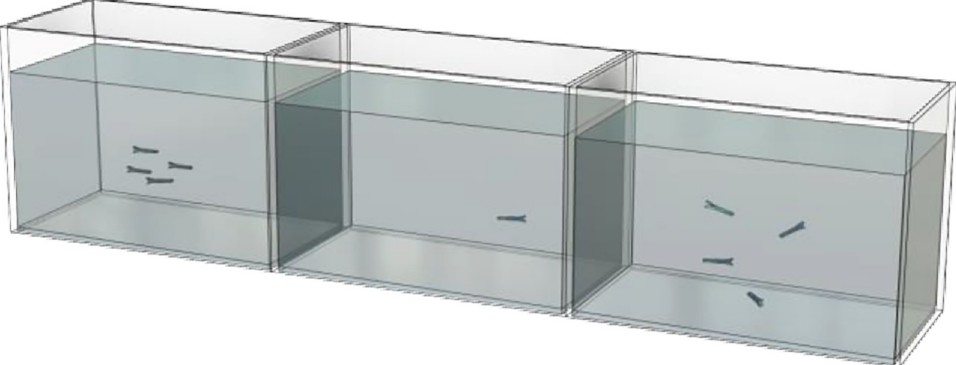

**Fig 1. Experimental apparatus.** Illustrative example with the subject in the center tank to test preference between an established shoal on the right and a newly-formed shoal on the left. The apparatus consists of three identical 18.9L (5.5 gal.) glass aquaria (each 20.3 cm wide × 25.4 cm high × 40.6 cm long; total width = 121.8 cm) filled with water to a depth of 21 cm.

sides of the center tank were covered with removable opaque barriers to obstruct the test subjects' vision into the flanking tanks. After the 3-minute habituation period, the partitions on either side of the central tank were removed; the rearmost wall remained opaque. During the subsequent 6-minute period of free swimming, EthoVision XT 15.0 recorded and tracked the subject's behavior.

Subject position within the central testing tank was recorded using a digital video camera mounted on a tripod positioned directly in front of the central testing tank. EthoVision XT 15.0 is a video tracking software program that can detect an animal in a live video feed, distinguish it from the background, and track the animal's movement, behavior, and activity. EthoVision XT 15.0 was programmed to analyze each recording in real-time, thus acquiring measurements of spatial location and subjects' proximity to adjacent tanks. Additionally, data were extracted offline to refine evaluation of swimming patterns and density. The tracking area within the central testing tank was divided into four quadrants (upper left, lower left, upper right, and lower right of the tank). Each quadrant of the arena constituted 25% of the total arena. Zebrafish preferences were characterized by swimming patterns as the program quantified time spent in each zone as well as zone transitions [56, 59]. Spatiotemporal data for each subject were captured and processed using EthoVision XT 15.0. Because of our study's focus on the subject's response to the level of intra-shoal familiarity, we ensured equal familiarity among the subjects of both stimulus shoals from the perspective of the subject fish to minimize bias. The method and layout of subject and shoal selection ensured that the subject was previously housed with only one fish of the novel shoal and only one fish of the established shoal.

## Design, measures, and analyses

This experiment was a 2 (Subject Sex) X 2 (Shoal Type) mixed-factorial design which was counterbalanced across both levels of presentation side for each shoal type (Novel-Left vs Established-Right or Established-Left vs Novel-right). Because no significant main effects or interactions were found for the Side factor, data were collapsed across Side for subsequent factorial analyses. The experimental design allowed for the experimental factors to be crossed against a flexibly-defined observation zone as an additional factor. As such, the analyses included four levels of quadrant (top left, bottom left, top right, bottom right) or three levels of vertical zone (left third, middle third, right third). While most statistical analyses used quadrants as the levels for the observational zone factor, certain analyses (e.g. side preference) were better examined using vertical zones as the levels of the observational zone factor. The following measures were obtained during the session with each subject:

- Cumulative duration percent within each quadrant: EthoVision quantified the total time each subject spent in each quadrant during the entire session, and the subsequent percentage of time within each quadrant was calculated for each subject's session.

- Percent of session time moving: EthoVision quantified the total duration of tracks recorded in each quadrant while the subject was moving at any velocity.

- Percent of session time freezing: EthoVision quantified the total duration of tracks recorded in each quadrant where the subject had ceased any detectable movement for a minimum of 3 seconds. In addition, swimming patterns were monitored by experimenters to ensure that no subject remained motionless for one minute or longer, which would necessitate discontinuation of the trial and exclusion of that subject's data (no subjects were excluded under this criterion).

- Average movement velocity during session: EthoVision measured velocity by dividing the distance the subject moved by the time difference between samples during motion tracking. EthoVision quantified the movement speed in mm/s for each subject's movement within each quadrant which was then used to calculate the average movement velocity within each quadrant during the session.

- Variability in Velocity: The variability in the velocity across tracking samples can be characterized by the Standard Deviation of the average movement velocity within each quadrant during the session.

- Number of transitions between quadrants: EthoVision quantified the number of transitions between adjacent quadrants (top right to bottom right & top left to bottom left as well as top right to top left & bottom right to bottom left). Similarly, EthoVision quantified the number of transitions between vertical zones (left, center, & right).

- Cross tank transitions: EthoVision quantified the number of transitions from the right side of the arena to the left side of the arena.

Using IBM SPSS (v.26), factorial data were analyzed using a Linear Mixed Model (LMM) with Type III Sums of Squares at α = .05. As heterogeneity of variance is common with these types of data [56], the model was set with a diagonal covariance structure and degrees of freedom for the denominator were adjusted using the Maximum Likelihood estimator for the LMM. Significant main effects and interactions were explored using unplanned comparisons with Bonferroni correction for family-wise error.

In order to characterize behavioral diversity across all zones in the observational arena, a single variable index based upon Shannon entropy [60] was calculated using the following formula:

$$H_n = -\sum p_i log_{10}(p_i) \tag{1}$$

Where $H_n$ is the index of behavioral diversity, $p_i$ is the proportion of cumulative session time spent in zone $i$, and $n$ is the total number of zones characterized with the index. The value of $H_n$ can range from 0 (only systematic variability) to 1.0 (completely random variability); higher values of $H_n$ indicate lower systematic variability in zone selection. This index has been used to characterize the movement of Humboldt penguins (*Spheniscus humboldti*) in a naturalistic zoo enclosure divided into zones of unequal sizes in order to examine the effects of live feeding events on the behavioral diversity of subjects across the enclosure [61]. Other studies have used this index to characterize the response of California blackworm (*Lumbriculus variegatus* to copper sulfate exposure [62] as well as the response of zebrafish and checker barbs (*Puntius oligolepis*) to different levels of structural complexity in artificial aquatic environments [63]. Therefore, $H_n$ is useful as a measure of characterizing systematic and random variability across a number of measures (e.g. duration in each zone) with a single index (for a review of behavioral diversity indices, see [64]). If the proportion of session time a subject in the current study spent in any particular zone is 1.0, then $H_n$ = 0.0. If a subject's proportionate time in all zones is equal across the zones, then $H_n$ = 1.0. Indices of $H_n$ between 0.0 and 1.0 indicate the extent to which a subject is systematically preferring any zone over the other zones.

## Results

Incomplete tracking data were obtained from two female subjects, and their respective data were excluded from analyses involving zone parameters. Tracking data from the remaining 80 subjects (45 males and 35 females) were included for analyses involving zone parameters.

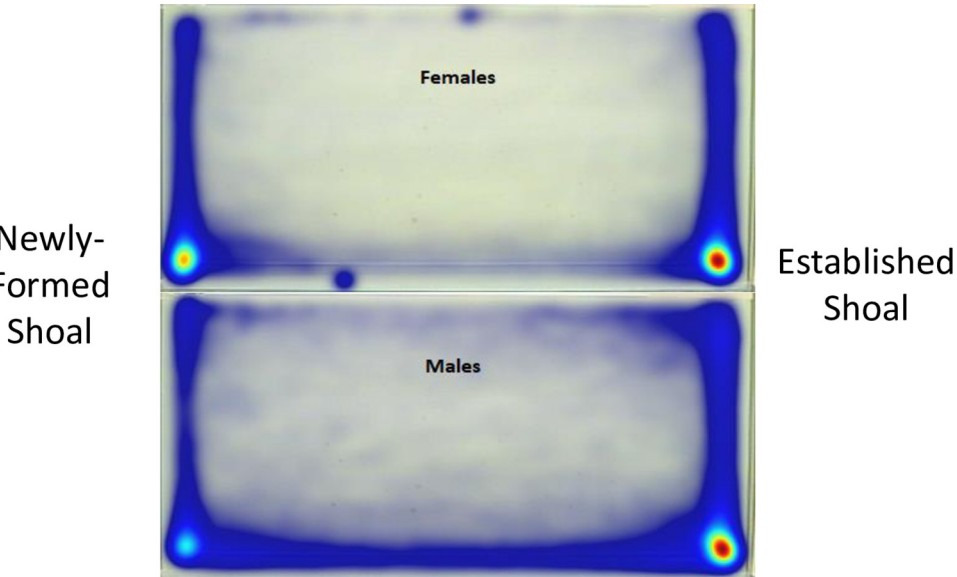

**Fig 2. Heatmaps collapsed across trials with the same configuration of stimuli.** Warm colors indicate highest intensities of localization while cooler colors indicate lowest intensities of localization.

## Duration within quadrants

Overall, subjects spent more time in lower zones adjacent to test shoals than in upper zones. Qualitatively, this difference in zone preference is demonstrated by location heat maps generated with Ethovision (see Fig 2). However, cumulative heat maps are limited in providing detail on the magnitude of these differences, which is better characterized using quantitative analyses. When analyzing the percent cumulative duration within quadrants, there was a significant main effect of Quadrant, $F$ (3,128.093) = 55.336, $p \leq$ .001; however, there was no significant main effect of Sex, $F$ (1,230.397) = 0.562, $p$ = .454, nor was there a significant interaction of Sex by Quadrant, $F$ (3,128.093) = 2.081, $p$ = .106 (Fig 3).

Across both sexes, subjects spent the most time in the lower quadrant closer to the established shoal ($M$ = 44.57, $SEM$ = 2.618), followed by the lower quadrant closer to the novel shoal ($M$ = 31.80, $SEM$ = 2.524). Subjects spent less time in the upper quadrant closer to the established shoal ($M$ = 15.69, $SEM$ = 1.272) and the least amount of time in the upper quadrant closer to the novel shoal ($M$ = 11.36, $SEM$ = 1.216) (Fig 3). The only pairwise comparison that is not significant is the difference between the time in the upper quadrant closer to the novel shoal and the upper quadrant closer to the established shoal (all other $p$'s $\leq$ .003).

## Time in motion

When analyzing the subjects' percent of session time in motion within quadrants, there was a significant main effect of Quadrant, $F$ (3,128.093) = 27.000, $p \leq$ .001; however, there was no significant main effect of Sex, $F$ (1,283.203) = 0.382, $p$ = .537, nor was there a significant interaction of Sex by Quadrant, $F$ (3,141.081) = 0.797, $p$ = .498 (Fig 4A).

Across both sexes, subjects spent the greatest percentage of time moving in the lower quadrant closer to the established shoal ($M$ = 9.91, $SEM$ = 0.562), followed by the lower quadrant closer to the novel shoal ($M$ = 8.00, $SEM$ = 0.397). Subjects spent less time moving in the upper quadrant closer to the established shoal ($M$ = 5.19, $SEM$ = 0.397) and the least amount of time moving in the upper quadrant closer to the novel shoal ($M$ = 4.24, $SEM$ = .417). The only

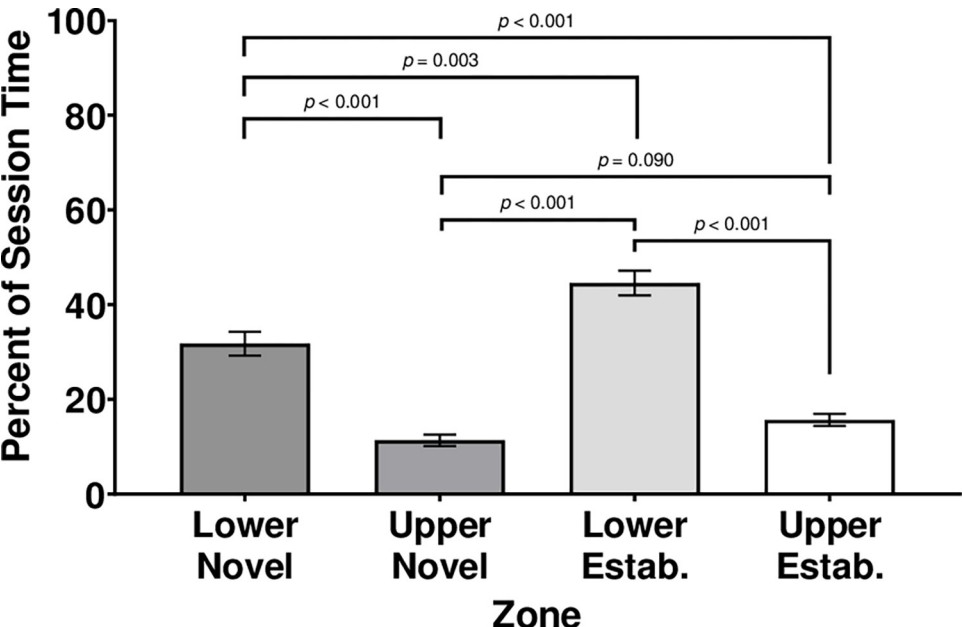

**Fig 3. Percent cumulative duration in each quadrant zone across both sexes.** Quadrants are labeled as upper/lower sections of the tank and novel/established stimulus shoal proximity. Error bars are +/- *SEM*. While exact *p*-values are reported for post-hoc comparisons, *p*-values smaller than 0.001 are indicated as *p* < 0.001.

pairwise comparisons that were not significant were the difference between the time in the upper quadrant closer to the novel shoal and the upper quadrant closer to the established shoal and the difference between the percent time moving in the lower quadrant closer to the novel shoal and the lower quadrant closer to the established shoal (all other *p*'s ≤ .001).

## Time motionless

When analyzing the percent of session time motionless within quadrants, there was a significant main effect of Quadrant, $F(3,124.260) = 58.281$, $p \leq .001$; however, there was no significant main effect of Sex, $F(1,215.332) = 1.26$, $p = .264$, nor was there a significant interaction of Sex by Quadrant, $F(3,124.260) = 2.625$, $p = .053$; (Fig 4B).

 Across both sexes, subjects spent the most time motionless in the lower quadrant closer to the established shoal ($M = 34.66$, $SEM = 2.192$), followed by the lower quadrant closer to the

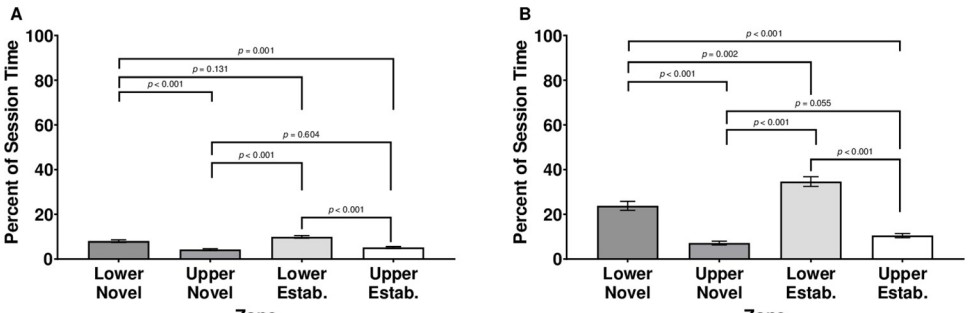

**Fig 4.** Percent duration in motion (A) and motionless (B) by zone across both sexes. Quadrants of the subject tank were labeled by upper/lower sections of the tank and novel/established stimulus shoal proximity. Error bars are +/- *SEM*. While exact *p*-values are reported for post-hoc comparisons, *p*-values smaller than 0.001 are indicated as *p* < 0.001.

novel shoal ($M$ = 23.79, $SEM$ = 2.524). Subjects spent less time in the upper quadrant closer to the established shoal ($M$ = 10.50, $SEM$ = 0.943) and the least amount of time in the upper quadrant closer to the novel shoal ($M$ = 7.12, $SEM$ = 0.865). The only pairwise comparison that is not significant is the difference between the time in the upper quadrant closer to the novel shoal and the upper quadrant closer to the established shoal (all other $p$'s ≤ .002).

### Average velocity

When analyzing the subjects' average velocity within quadrants, there were significant main effects of Quadrant, $F$ (3,180.743) = 22.925, $p$ ≤ .001, and Sex, $F$ (1,282.043) = 6.398, $p$ = .012 (Fig 5A); however, there was no significant interaction of Sex by Quadrant, $F$ (3,180.743) = 0.582, $p$ = .628; Females ($M$ = 60.26 mm/s, $SEM$ = 2.503) had a faster average velocity than males ($M$ = 51.84 mm/s, $SEM$ = 2.195).

Across both sexes, subjects moved at the slowest velocity in the lower quadrant closer to the established shoal ($M$ = 40.34 mm/s, $SEM$ = 2.342), followed by the lower quadrant closer to the novel shoal ($M$ = 50.52, $SEM$ = 4.028). Subjects swam faster in the upper quadrant closer to the established shoal ($M$ = 61.20 mm/s, $SEM$ = 3.413) and swam the fastest in the upper quadrant closer to the novel shoal ($M$ = 72.15 mm/s, $SEM$ = 3.313). The only pairwise comparisons that were not significant were the difference between the average velocity in the upper quadrant closer to the novel shoal and the upper quadrant closer to the established shoal and the difference between the average velocity in the upper quadrant closer to the established shoal and the lower quadrant closer to the novel shoal (all other $p$'s ≤ .001).

### Variability in velocity

One measure of variability in velocity is the standard deviation of each subject's movement velocity within each quadrant. When analyzing the subjects' standard deviation of velocity within quadrants, there was a significant main effect of Quadrant, $F$ (3,160.255) = 25.926, $p$ ≤ .001; however, there was no significant main effect of Sex, $F$ (1,296.650) = 0.38, $p$ = .847, nor was there a significant interaction of Sex by Quadrant, $F$ (3,160.255) = 1.291, $p$ = .279 (Fig 5B).

Across both sexes, subjects had the lowest variability in velocity while swimming in the lower quadrant closer to the established shoal ($M$ = 64.70 mm/s, $SEM$ = 1.213), followed by the lower quadrant closer to the novel shoal ($M$ = 68.57 mm/s, $SEM$ = 1.581). Subjects had greater variability in their movement velocities in the upper quadrant closer to the established shoal ($M$ = 73.99 mm/s, $SEM$ = 1.325) and the highest variability in the upper quadrant closer to the novel shoal ($M$ = 78.71 mm/s, $SEM$ = 1.140). The only pairwise comparisons that were not

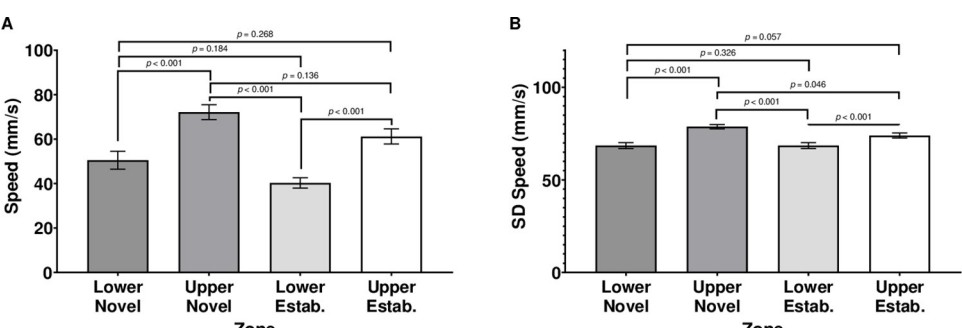

**Fig 5.** Average velocity (A) and velocity SD (B) during movement. Quadrants of the subject tank were labeled by upper/lower sections of the tank and novel/established stimulus shoal proximity. Error bars are +/- $SEM$. While exact $p$-values are reported for post-hoc comparisons, $p$-values smaller than 0.001 are indicated as $p$ < 0.001.

significant were the differences between variability in movement velocities in upper quadrant closer to the established shoal and the lower quadrant closer to the novel shoal and the difference between the average variability of subjects' movement velocities in the lower quadrant closer to the established shoal and the lower quadrant closer to the novel shoal (all other $p$'s ≤ .046).

## Zone and cross-tank transitions

Inter-quadrant transitions were analyzed by a 4 (Quadrant of Origin) by 2 (Sex) LMM. There was a significant main effect for Sex, $F$ (1, 312.96) = 4.63, $p$ = 0.032. There was neither a significant main effect for Quadrant of Origin, $F$ (1, 152.04) = 0.996, $p$ = 0.397, nor a significant interaction of Quadrant of Origin by Sex, $F$ (1, 152.04) = 0.236, $p$ = 0.871 (Fig 6). Across all Quadrants of Origin, females ($M$ = 15.55, $SEM$ = 1.064) had significantly more quadrant-to-quadrant transitions than males ($M$ = 12.51, $SEM$ = 0.933). To assess cross-tank transitions, data were collapsed across upper and lower quadrants, and an independent-samples $t$-test was performed between males and females on cross-tank transitions. Males ($M$ = 14.33, $SEM$ = 1.253) had significantly more cross-tank transitions than females ($M$ = 7.73, $SEM$ = 1.053), $t$ (80) = 3.930, $p$ ≤ .001.

## Behavioral diversity index

Sufficient data were obtained from the two excluded females to be included in the behavioral diversity index measures, providing a total sample size of 82 subjects (34 males and 37 females). For the current data, two indices of $H_n$ were calculated for each subject. One index, $H_4$, considered the four quadrants (upper right, upper left, lower right, lower left) of the observation arena while the other index, $H_3$, considered the left, middle, and right vertical thirds of the observation arena. $H_4$ thus provides an index which characterizes diversity in zone selection both vertically and horizontally within the observation arena, while $H_3$ provides an index

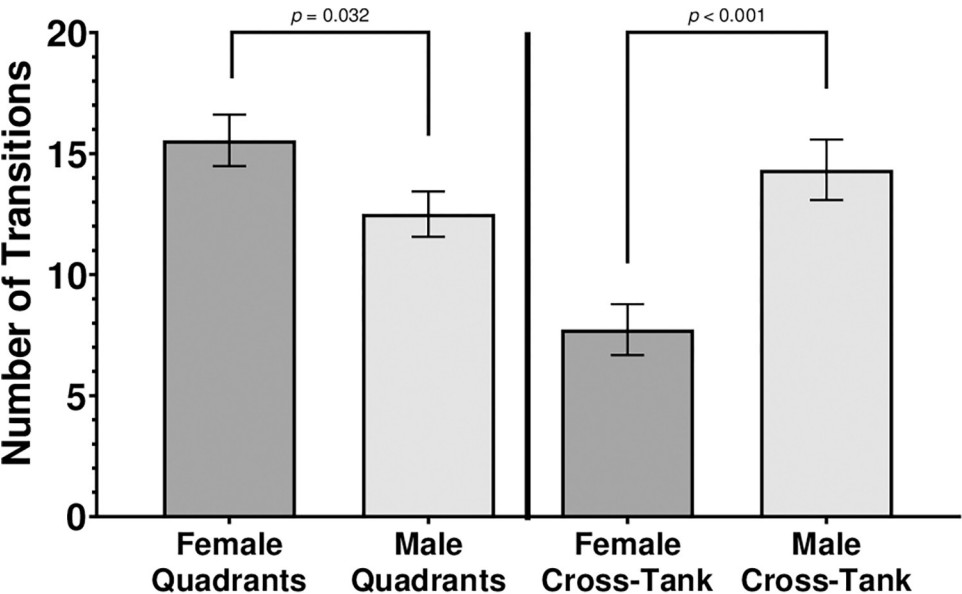

**Fig 6. Sex differences in inter-quadrant and cross-tank transitions.** Quadrants of the subject tank were labeled by upper/lower sections of the tank and novel/established stimulus shoal proximity. Error bars are +/- SEM. While exact $p$-values are reported for post-hoc comparisons, $p$-values smaller than 0.001 are indicated as $p < 0.001$.

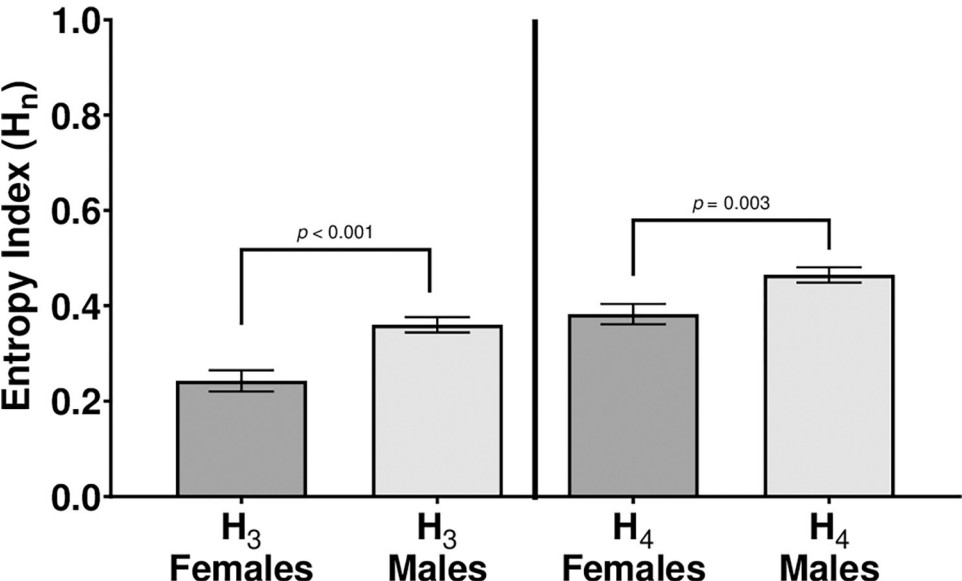

**Fig 7. Sex differences in behavioral entropy.** $H_3$ measures entropy across three vertical zones while $H_4$ measures entropy across four quadrants. Error bars are +/- *SEM*. While exact *p*-values are reported for post-hoc comparisons, *p*-values smaller than 0.001 are indicated as $p < 0.001$.

which characterizes diversity in zone selection horizontally across the observation arena. Higher values of either $H_4$ or $H_3$ indicate more random variability in a subject's proportions in zone utilizations. Whenever a case for a subject had a proportion for any zone equal to zero, resulting in an undefined computation for the log of that value, the undefined result was set to zero and the summation for $H_n$ continued across the other non-zero proportions.

As expected, males had significantly higher behavioral entropy index scores than females for $H_3$ analyzed across three zones (males $M = 0.360$, $SEM = 0.016$; females $M = 0.243$, $SEM = 0.022$; $t(80) = 4.365$, $p \leq .001$, one-tailed). Males also had significantly higher behavioral entropy index scores than females for $H_4$ analyzed across four zones, (males $M = 0.465$, $SEM = 0.016$; females $M = 0.383$, $SEM = 0.021$; $t(80) = 3.12$, $p = 0.033$, one-tailed) (Fig 7). Taken together, these results indicate that male subjects show less systematic variability in their utilization of zones in the observation arena than female subjects, revealing sex differences in overall responding not revealed by main effects or interactions in the Sex X Zone factorial analyses of the specific dependent measures (except for the main effect of Sex on swim velocity).

## Discussion

The results presented here indicate that solitary zebrafish can differentiate between established and novel shoals based solely on visual cues and choose to spend more time near an established shoal. Movement measures indicate that both sexes spent more time motionless when in the lower quadrant nearest the established shoal. Velocity measures show that both sexes exhibited less variability in velocity when in the upper quadrant near the established shoal compared to the upper quadrant near the novel shoal, potentially indicative of less darting behavior. While both male and female zebrafish spent more time in the lower quadrant near the established shoal, there were significant differences in the manner by which males and females moved around the field; females appeared to investigate more locally while males investigated more across the two shoals. Use of the Shannon entropy measure further confirmed the increased behavioral entropy for males and provides a useful measure for future comparison across

studies that use different methods and arenas. The addition of shoal stimuli of varying familiarity to the three-tank open-swim preference test provides an attractive and easy-to-use system for studies investigating zebrafish models of disorders that affect social behavior and recognition of social cues.

The present analyses include a novel application of Shannon entropy to characterize the diversity in zone preference in the three-tank open-swim preference test. The three-tank open-swim preference test has been used successfully in a number of previous studies, and it is becoming increasingly popular for the investigation of social preference in both normal populations (e.g. EKK or AB wild type) [26, 56, 65] and in clinical models [11, 61]. Previous studies established the efficacy of the technique and demonstrated experimental effects through the analysis of unitary measures such as duration in zone, swimming velocity, etc. The present study establishes the utility of a unitless behavioral diversity index to characterize the extent to which subjects demonstrate a systematic preference among the observational zones of interest in the three-tank open-swim preference task and allows for comparison of the diversity index across different zone characterizations (e.g. quadrants versus vertical thirds) and between subject groupings (e.g. sex). As the nature of the three-tank open-swim preference test is to determine the extent to which test subjects demonstrate a preference and/or avoidance of zones in proximity with test stimuli, the application of an index of behavioral diversity such as Shannon entropy can be useful for future studies of social preference in tasks involving movement in open arenas.

Previous studies have demonstrated sex differences in zebrafish preference based on number of individuals in a shoal [66] and pigment patterns [67]. Our results suggest sex-specific differences in zebrafish preference when given a choice between a newly-formed shoal and an established shoal, with females exhibiting less entropy, a greater preference for the established shoal and increased average swimming speed. Swimming speed and vertical tank location during exposure to novel environments and other common stressors has been linked to anxiety levels in zebrafish [68, 69]. Increased swimming speeds could indicate darting patterns that result from an expression of fear [70], while more time spent in the lower half of a novel tank is indicative of an anxious state which can be reversed with exposure to anxiolytic drugs [71]. Both male and female zebrafish exhibit differences in anxiety-like responses with females spending more time in the bottom half of the tank during a novel tank task and more time in the dark zone during a light-dark task [72]. Further, experiments on wild-caught zebrafish show that males are bolder during feeding than females [73]. Our results show a greater percentage of time spent in the lower quadrants for both sexes, indicating an anxious state for both male and female subjects. However, the faster average swimming speeds of female subjects observed in this study could be indicative of a higher level of anxiety compared to male subjects, suggesting that the social choice paradigm can elicit subtle behavioral anxiety differences compared to the novel tank task. Further, anxiety level has the potential to induce shoaling behavior, as it is exhibitive of seeking out energy rationing [74]. The subtle behavioral differences exhibited by females in our study (lower entropy and higher tendency to prefer the established shoal) is potentially due to higher anxiety levels.

Zebrafish place preference can be influenced by chemical [70, 75] or live stimuli [56, 76], and a three-chamber apparatus is commonly utilized to quantify preference depending on subject swimming patterns [11, 56]. As an example of place bias indicating social choice, Social Preference Index has been utilized in several studies concerning social preference in zebrafish, equating proximity with preference for interaction [77–79]. Swimming patterns localized near one stimulus over another could indicate preference, but other driving forces of social behavior, such as investigative or aggressive interactions, should not be discounted. The present study solely investigated social interaction based on visual information and did not delineate

between aggression versus investigation. However, aggressive interaction or investigation would also indicate a change in social interaction that occurs due to the observed differences in stimuli shoal cohesion, suggesting that the subject fish were still able to differentiate between shoal types.

A salient component of Fetal Alcohol Spectrum Disorders (FASD), Autism Spectrum Disorders (ASD), and several other neurodevelopmental disorders' symptomatology is the display of atypical social behavior. Since the zebrafish model exists at the intersection of behavioral complexity and biological simplicity, use of the zebrafish to study neurodevelopmental disorders has recently gained popularity. Shoaling tendencies are first distinguishably exhibited by young zebrafish approximately two weeks post-hatch, and shoaling as a species-specific behavior is critically influenced by early life experience [58]. Zebrafish exposed to alcohol at the embryonic stage have been shown to display impaired shoaling behavior development [80]. In humans, FASD is a life-long disorder, and social impairments including social withdrawal and depression persist for the duration of a patient's life. Similarly, embryonic zebrafish exposed to ethanol exhibit severely reduced shoaling responses that continue two years following initial exposure and derive from central nervous system changes rather than motor or visual dysfunction [80, 81]. The established versus new shoaling model outlined here contributes to understanding social behavior in typically-developing zebrafish, though the model may be useful to better classify social impairment in neurodevelopmental disorders such as FASD or ASD. Social interaction difficulties in FASD and ASD may be evident through a failure to recognize differences in established versus new shoals. Such behavior would be supported by a lack of preference for either shoal, shown by a subject spending equal amounts of time in close proximity to either stimuli.

The DYRK1A gene, located in the "Down Syndrome Critical Region DSCR" has been identified as a significant element in the pathogenesis of ASD in humans [82]. DYRK1A mutation in humans is connected to intellectual impediments, microcephaly, and ASD. When DYRK1A is knocked-out in zebrafish (DYRK1A KO), affected subjects exhibit social abnormalities parallel to those displayed by human ASD patients. Specifically, DYRK1A KO resulted in decreased expression of c-fos, a proto-oncogene important for cellular proliferation and differentiation [82, 83]. When presented with a three-member social stimuli shoal, DYRK1A KO zebrafish spent significantly less time in the zone of closest proximity to the shoal compared to wildtype (WT) zebrafish [82]. Using the DYRK1A KO in our three-chamber social choice model with established versus newly-formed shoals as flanking stimuli would determine if DYRK1A expression is necessary for identifying intragroup familiarity between shoals.

Our findings and conclusions present implications for both basic research on the mechanisms of social preference in animals as well as the aforementioned zebrafish models of human disease and behavioral dysfunction. Future research on social preference using the three-tank open-swim preference test could explore the various characteristics of intra-shoal activity that indicate social novelty within a newly-established shoal which are subsequently detected by the observing subject. In addition, future research could also explore whether other factors such as age, size, or health status affect preference for an established shoal, and whether other features of test shoals and/or individual subjects can override the preference for established shoals. In order to extend and further explore various aspects of subjects' movement, future researchers could use a more-sophisticated dual-camera setup that allows for analysis of three-dimensional movement data [84, 85] which may subsequently reveal other, perhaps more subtle, differences in shoal preference. Finally, the demonstrated value of Shannon entropy in the current study introduces new possibilities for the comparison of arena-based movement of subjects under a variety of conditions across studies.

## Supporting information

**S1 Data. Novel-ExpShoalSSP_PlosONE.**
(XLSX)

## Acknowledgments

The authors thank John Velkey of the University of Maryland–Baltimore County for his assistance with figures and Jake Boles of the University of Florida for technical assistance with the use of EthoVision XT 15.0.

## Author Contributions

**Conceptualization:** Andrew J. Velkey, Caroline H. Koon, Katie M. Wiens.

**Data curation:** Andrew J. Velkey, Caroline H. Koon, Isabel A. Danstrom.

**Formal analysis:** Andrew J. Velkey.

**Funding acquisition:** Andrew J. Velkey, Caroline H. Koon.

**Investigation:** Andrew J. Velkey, Caroline H. Koon, Isabel A. Danstrom.

**Methodology:** Andrew J. Velkey, Caroline H. Koon, Isabel A. Danstrom.

**Project administration:** Andrew J. Velkey, Caroline H. Koon, Isabel A. Danstrom.

**Resources:** Andrew J. Velkey.

**Software:** Caroline H. Koon, Isabel A. Danstrom.

**Supervision:** Andrew J. Velkey, Katie M. Wiens.

**Validation:** Caroline H. Koon, Isabel A. Danstrom.

**Visualization:** Andrew J. Velkey, Katie M. Wiens.

**Writing – original draft:** Andrew J. Velkey, Caroline H. Koon, Isabel A. Danstrom, Katie M. Wiens.

**Writing – review & editing:** Andrew J. Velkey, Caroline H. Koon, Isabel A. Danstrom, Katie M. Wiens.

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
