## [Decision Letter · Decision Letter 0]

12 May 2022

PONE-D-22-06149Female Zebrafish (Danio rerio) Demonstrate Stronger Preference for Established Shoals over Newly-Formed Shoals in the Three-Tank Open-Swim Preference TestPLOS ONE

Dear Dr. Velkey,

Thank you for submitting your manuscript to PLOS ONE.Your paper has been revised by two reviewers expert in the field and aAfter careful consideration, we feel that it has merit but does not fully meet PLOS ONE’s publication criteria as it currently stands. Therefore, we invite you to submit a revised version of the manuscript that addresses the points raised during the review process.

We look forward to receiving your revised manuscript.

Kind regards,

Cristiano Bertolucci, Ph.D.

Academic Editor

PLOS ONE

Journal Requirements:

Reviewers' comments:

Reviewer's Responses to Questions

**Comments to the Author**

1. Is the manuscript technically sound, and do the data support the conclusions?

Reviewer #1: Partly

Reviewer #2: Yes

2. Has the statistical analysis been performed appropriately and rigorously? 

Reviewer #1: Yes

Reviewer #2: Yes

3. Have the authors made all data underlying the findings in their manuscript fully available?

Reviewer #1: Yes

Reviewer #2: Yes

4. Is the manuscript presented in an intelligible fashion and written in standard English?

Reviewer #1: Yes

Reviewer #2: Yes

5. Review Comments to the Author

Reviewer #1: The study examines the shoaling preference of zebrafish and their tendency to spend a greater time with established shoals in comparison to newly formed shoals. The authors have used an open swim preference test to address this question. The study is interesting and provides support for existing earlier observation on the shoaling preference of zebrafish for familiar individuals instead of unfamiliar ones. While the results are not necessarily novel, the authors have used a number of parameters to assess association preferences and provide a measurement based on Shannon entropy to quantify transition differences across the test tank to indicate association tendencies towards established shoals. While this method has been used in some studies earlier that uses this index for measuring activity levels under stress conditions in organisms, its application as a measure for differences in transition across tank when presented with shoal choice is also demonstrated in this study. See these papers for relevant literature that use the index:

Ji, C. W., Park, Y. S., Cui, Y., Wang, H., Kwak, I. S., & Chon, T. S. (2020). Analyzing the response behavior of Lumbriculus variegatus (Oligochaeta: Lumbriculidae) to different concentrations of copper sulfate based on line body shape detection and a recurrent self-organizing map. International Journal of Environmental Research and Public Health, 17(8), 2627.

Kistler, C., Hegglin, D., Würbel, H., & König, B. (2011). Preference for structured environment in zebrafish (Danio rerio) and checker barbs (Puntius oligolepis). Applied Animal Behaviour Science, 135(4), 318-327.

While the manuscript is well written, there are sections that seem somewhat redundant and need to be shortened, especially in the Introduction. Specific comments on these are attached separately with the manuscript, as a pdf. Methodology needs to be more detailed and provide information on how the stimuli shoal and test subjects were set.. was there any chance of prior familiarity between the stimuli an the test subjects? What was the sample size used for the test subjects? Were same stimuli shoals reused for the trials?

The study suggests that compared to males, females show a stronger preference for established shoals for most of the studied parameters. The authors need to discuss this aspect in some more detail. Is there prior literature on sex specific difference in preferences among fish? In what context do we see them? In the present study this difference is suggested to be a result of stress. Why do female individuals show greater levels of stress?

Specific comments are attached in the pdf.

Reviewer #2: The study investigated sex differences and preference for newly formed versus established shoals in zebrafish. A big and interesting amount of data is provided, but some issues need to be better explained and clarified.

- There are discrepancies concerning size and volume of the test tank (15.2 Vs. 15.3 cm, 7.6 L Vs. 9.5 L) comparing methods and figure 1 legend.

- The authors mention 4 separate holding tanks and 8 separate same-sex groups. This is confusing and should be clarified. How many tanks, which size, fish density, acclimation period? How were stimulus shoals kept compared to test subjects?

- Was the water changed in the experimental apparatus between test subjects?

- Entropy formula is missing a minus sign (so that negative results become positive).

- It is advisable to use exact p-values for the post hoc comparisons in the graphs.

- Please adjust the y-axes of plots in figure 4 so that they better represent the outcome illustrated (as they are, y-axes are identical to figure 3, thought the variables are different).

- I wonder whether the analysis presented in figure 4 (time in motion and motionless per quadrant) is somewhat futile considering the results for percent time spent per quadrant. If a fish spends more time on a given quadrant, this time is spent either in motion or motionless, so results follow the same pattern. I think the hypothesis behind the investigation of these outcomes as they were presented should be clearly stated. Perhaps the more relevant analyses here would be percent time spent in motion or motionless corrected for the total time spent in the quadrant. This would reveal whether movement was increased or decreased in the most preferred zones. Without adjusting for time in the zone, the rationale is not clear.

- Figure legends mention (a) and (b) in some cases, but such labeling is not presented in the figure.

- Please keep variable names the same throughout the manuscript and plots (e.g., average velocity Vs. speed Vs. swim speed).

- Does EthoVision discount motionless time to calculate average velocity or is distance moved divided by the total time spent in the quadrant?

- It is not clear how a standard deviation for a single subject was calculated considering that fish were tested only once. What are exactly the data points for a given subject that were used to calculate variability in velocity per quadrant?

- Considering there is a main effect of sex for average velocity, why show only collapsed results?

- Sample size is not clear; total number of animals used include stimulus shoals? Please clarify.

- The authors state that animals motionless for more than one minute were excluded. How many animals were excluded by this criterion? Proportion of males Vs. females equal? Were any other criteria for excluding animals employed?

- Were males and females randomly selected for outcome assessment? Please describe whether any measures to counterbalance sex in the order of behavioral testing were taken or not. Also, were the animals randomly housed during the experiment?

- What is the rationale for the chosen sample size? Did the authors perform a priori power calculations?

- Plots: please use dot plots instead and show the error bars for both directions (10.1371/journal.pbio.1002128). It is also advisable to use standard deviation instead of standard error of the mean.

- Supplementary material cannot be understood on its own. A variable key should be provided. For example, the reader cannot guess whether 1 is the code for males or for females, and the difference between columns Q (InzoneArenaCenterpointCumulativeDurations) and R (InzoneArenaCenterpointCumulativeDuration) is not very obvious.

6. PLOS authors have the option to publish the peer review history of their article (what does this mean?). If published, this will include your full peer review and any attached files.

Reviewer #1: No

Reviewer #2: No

---

## [Author Response · Author response to Decision Letter 0]

26 Jul 2022

Reviewer 1, Comments in Editorial Response Email of 12 May, 2002

SUMMARY: The study examines the shoaling preference of zebrafish and their tendency to spend a greater time with established shoals in comparison to newly formed shoals. The authors have used an open swim preference test to address this question. The study is interesting and provides support for existing earlier observation on the shoaling preference of zebrafish for familiar individuals instead of unfamiliar ones…. While the manuscript is well written, there are sections that seem somewhat redundant and need to be shortened, especially in the Introduction. Specific comments on these are attached separately with the manuscript, as a pdf (addressed in the next section)

While the results are not necessarily novel, the authors have used a number of parameters to assess association preferences and provide a measurement based on Shannon entropy to quantify transition differences across the test tank to indicate association tendencies towards established shoals. While this method has been used in some studies earlier that uses this index for measuring activity levels under stress conditions in organisms, its application as a measure for differences in transition across tank when presented with shoal choice is also demonstrated in this study. See these papers for relevant literature that use the index:

Ji, C. W., Park, Y. S., Cui, Y., Wang, H., Kwak, I. S., & Chon, T. S. (2020). Analyzing the response behavior of Lumbriculus variegatus (Oligochaeta: Lumbriculidae) to different concentrations of copper sulfate based on line body shape detection and a recurrent self-organizing map. International Journal of Environmental Research and Public Health, 17(8), 2627.

Kistler, C., Hegglin, D., Würbel, H., & König, B. (2011). Preference for structured environment in zebrafish (Danio rerio) and checker barbs (Puntius oligolepis). Applied Animal Behaviour Science, 135(4), 318-327.

-We have added the citations for these two studies to this section of the manuscript. 

Methodology needs to be more detailed and provide information on how the stimuli shoal and test subjects were set.. was there any chance of prior familiarity between the stimuli an [sic] the test subjects? 

-We have clarified how stimuli fish and subjects were selected, and we have added specific information regarding prior familiarity between subjects and test stimuli in the Materials, apparatus, and procedure subsection.

What was the sample size used for the test subjects? 

-This information is included in the Subjects subsection, and we have added a clarification to indicate that additional fish used as visual stimuli are not included in the sample size. 

Were same stimuli shoals reused for the trials?

-Shoal stimuli reuse is clarified in the Materials, apparatus, and procedure subsection.

The study suggests that compared to males, females show a stronger preference for established shoals for most of the studied parameters. The authors need to discuss this aspect in some more detail. Is there prior literature on sex specific difference in preferences among fish? In what context do we see them? In the present study this difference is suggested to be a result of stress. Why do female individuals show greater levels of stress?

-We appreciate this suggestion, and have modified the third paragraph in the Discussion section to address this particular issue. 

Reviewer 1, Comments in PDF of submission:

Page 9: There seems a lot of repetition between this paragraph and the previous one. Please remove redundant statement to improve readability.

-We have deleted redundant statements from the paragraph.

Page 10: This is not entirely true. Please see:

Ogi, A., Licitra, R., Naef, V., Marchese, M., Fronte, B., Gazzano, A., & Santorelli, F. M. (2021). Social preference tests in zebrafish: A systematic review. Frontiers in Veterinary Science, 7, 1239.

-While Ogi et al. review studies that looked at kinship and familiarity of subjects to shoal, we did not see any study looking at subject choice based on the level of social familiarity within the shoal. However, we have modified this statement to reflect "less data" as opposed to "not known".

Page 10: There are studies that examine role of kin and familiarity in shoal choice. These might be useful and relevant:

Frommen, J. G., Mehlis, M., Brendler, C., & Bakker, T. (2007). Shoaling decisions in three-spined sticklebacks (Gasterosteus aculeatus)—familiarity, kinship and inbreeding. Behavioral Ecology and Sociobiology, 61(4), 533-539.

Cattelan, S., & Griggio, M. (2020). Within-shoal phenotypic homogeneity overrides familiarity in a social fish. Behavioral Ecology and Sociobiology, 74(4), 1-8.

-These two studies concern preference for familiars based on subject:shoal familiarity. We have already summarized similar studies and feel the addition of these citations within the suggested paragraphs would be redundant and potentially confusing to the reader. Therefore, we have decided to leave the section as originally submitted and will not be including these references. 

Page 10, Line 74: provide relevant literature to support this statement.

-We have simplified this sentence and added the following citation in support of this statement: 

McRobert, S. P., & Bradner, J. (1998). The influence of body coloration on shoaling preferences in fish. Animal Behaviour, 56(3), 611-615.

Page 10: This section can be shortened. This again has a lot of redundant statements. 

-This section concerns methodology and procedure. Any repetition in the presentation of this information is used to ensure clarity in our description.

Page 11: Shape? What does this imply?

-We have changed this to “collective group shape” to better describe the characteristic of interest here. 

Page 14: What do you mean by "investigational" lines? Replace this with a better term.

-We have changed the term to “transgenic lines” to clarify.

Page 14: Providing information of your results seems out of place in the Introduction. Instead of presenting the summary of your results here, it is important to outline the broad aims and predictions of the study. 

-Many authors choose to summarize their findings in the final paragraph of the introduction in order to help the reader more easily interpret the upcoming results, which are extensive. We have left the summary in place.

Page 15: Why was 7 days established as the time for development of intrashoal familiarity? Have there been studies that suggest this?

-We have added a reference to support our selection of the 7-day period:

Buske, C., & Gerlai, R. (2011). Shoaling develops with age in Zebrafish (Danio rerio). Progress in Neuro-Psychopharmacology and Biological Psychiatry, 35(6), 1409-1415.

Page 15: Where were the fish for the established shoal taken from? Did you ensure the subject fish were not familiar with individuals of the established shoal? 

AND

Page 15: Where were the experimental subjects selected from? Were they the same individuals that were kept in the holding tanks? How did you ensure that this did not lead to bias with respect to possible familiarity with members kept in the same holding tanks?

-We have clarified how stimuli fish and subjects were selected, and we have added specific information regarding prior familiarity between subjects and test stimuli in the Materials, apparatus, and procedure subsection. We controlled for any potential bias by selecting one member of each shoal stimulus from the same holding tank as the test subject so that any potential familiarity between the subject and members of each shoal stimulus was distributed between both shoal stimuli. Visual barriers were placed between all holding tanks, so fish in one holding tank were not able to establish visual familiarity with fish in the other holding tanks.

Page 16: What do you mean by "in-sex"? Do you mean same sex? Please reword.

-We have changed “in-sex” to “same-sex” to clarify the statement.

Page 16: What was the final sample size of the subject fish for each set of choice experiments? Provide all details of numbers of stimuli fish used, and whether they were reused for tests or used once per choice test.

-We have clearly stated the total sample size, including the number of males and the number of females, in the Subjects subsection, and we have clarified the reuse of fish as stimuli in the Materials, apparatus, and procedure subsection.

Page 20: Please include the sample size of subjects used for the analysis

-We have clearly re-stated the total sample size, including the number of males and the number of females, and indicated the exclusion of data from two female subjects from most of the analyses in the opening of the Results section.

Reviewer 2, Comments in Editorial Response Email of 12 May, 2002

There are discrepancies concerning size and volume of the test tank (15.2 Vs. 15.3 cm, 7.6 L Vs. 9.5 L) comparing methods and figure 1 legend.

-The errors in the description of physical apparatus for housing and experimental procedures have been corrected. We appreciate the reviewer's attention to detail regarding our error.

The authors mention 4 separate holding tanks and 8 separate same-sex groups. This is confusing and should be clarified. How many tanks, which size, fish density, acclimation period? How were stimulus shoals kept compared to test subjects?

-We have clarified the description of housing, acclimation, etc. in the Materials, apparatus, and procedure subsection.

Was the water changed in the experimental apparatus between test subjects?

-Information regarding 50% weekly water changes has been included in the Materials, apparatus, and procedure subsection. 

Entropy formula is missing a minus sign (so that negative results become positive).

-We have corrected this error.

It is advisable to use exact p-values for the post hoc comparisons in the graphs.

-We have included this change in all of the graphs, with extremely small p-values represented as “p < 0.001”

Please adjust the y-axes of plots in figure 4 so that they better represent the outcome illustrated (as they are, y-axes are identical to figure 3, thought the variables are different).

-We appreciate the reviewer’s suggestion here regarding the visualization of the specific results. However, we have decided to keep the y-axes in figs 3 and 4 the same as the same unit of measurement "Percent of Session Time" on the y-axes, is used for all three DV's. When y-axes are used to represent percentages / proportions, we consider it best practice to use the full range on the y-axis to prevent visual distortion of results. Furthermore, keeping the y-axes identical here allows the reader to compare results across the DV’s.

I wonder whether the analysis presented in figure 4 (time in motion and motionless per quadrant) is somewhat futile considering the results for percent time spent per quadrant. If a fish spends more time on a given quadrant, this time is spent either in motion or motionless, so results follow the same pattern. I think the hypothesis behind the investigation of these outcomes as they were presented should be clearly stated. Perhaps the more relevant analyses here would be percent time spent in motion or motionless corrected for the total time spent in the quadrant. This would reveal whether movement was increased or decreased in the most preferred zones. Without adjusting for time in the zone, the rationale is not clear.

-We appreciate the reviewer’s perspective regarding these particular dependent variables and our analytic strategy, but we do not consider these analyses to be “futile”. The selection of these particular dependent variables and analyses is consistent with reports by previous investigators (Velkey et al, 2019; Speedie & Gerlai, 2008, etc.). Moreover, one of the key points in our current report is that unit-based measures, whether obtained as data or derived as secondary measures from the obtained data (as the reviewer suggests above), may not reveal differences of interest to investigators. Our use of Shannon entropy as a unit-less measure reveals sex differences in responding that are not, with the exception of one dependent variable, revealed through careful analysis of unit-based measures. As the selection of these particular dependent variables is consistent with those of previous reports, we have elected to retain their inclusion in the current report.

Figure legends mention (a) and (b) in some cases, but such labeling is not presented in the figure.

-We have corrected the error and added the labels where needed.

Please keep variable names the same throughout the manuscript and plots (e.g., average velocity Vs. speed Vs. swim speed).

-We have changed the term “swim speed” to “movement velocity” throughout the revised manuscript to maintain consistency.

Does EthoVision discount motionless time to calculate average velocity or is distance moved divided by the total time spent in the quadrant?

-According to the Ethovision 10.0 User Manual, “Velocity is obtained by dividing the Distance moved by the time difference between a sample and the previous one”, and Distance is defined as, “The distance traveled by the center, nose, or tail-base point of the subject from the previous sample to the current one” (pp. 572-2). As such, EthoVision does not exclude motionless time in the calculations for average velocity or average distance. When the subject stops moving for any period of time, the Distance moved and subsequent Velocity measure for those samples is equal to a true zero. As such, the average velocity should include these zero measures as we have done in the current study.

It is not clear how a standard deviation for a single subject was calculated considering that fish were tested only once. What are exactly the data points for a given subject that were used to calculate variability in velocity per quadrant?

-The operational definitions for movement velocity and variability in movement velocity are clarified in the Design, measures, and analyses subsection.

Considering there is a main effect of sex for average velocity, why show only collapsed results?

-Figure 5A represents the main effects of quadrant on average swimming speeds. As there is no Sex by Quadrant interaction, the results are collapsed across sex to illustrate the effect of interest. No illustration is necessary for the simple main effect of sex.

Sample size is not clear; total number of animals used include stimulus shoals? Please clarify.

-We have clearly stated the total sample size, including the number of males and the number of females, in the Subjects subsection

The authors state that animals motionless for more than one minute were excluded. How many animals were excluded by this criterion? Proportion of males Vs. females equal? Were any other criteria for excluding animals employed?

-The wording in this section is clarified to indicated that no subjects were excluded due to this criterion.

Were males and females randomly selected for outcome assessment? Please describe whether any measures to counterbalance sex in the order of behavioral testing were taken or not. Also, were the animals randomly housed during the experiment?

-We clearly indicate that subjects were selected randomly from home tanks in the Materials, apparatus, and procedure subsection. We did not randomize the order in which males and females were selected for testing. Counterbalancing is typically used as an experimental control over potential carryover of a within-subjects treatment. As subject sex is a between-subjects factor, carryover of any sex-order effect is not possible. Perhaps the reviewer is concerned that there is some potential for confounding of testing order, but we fail to see how this is a potential issue for the current study. We addressed housing and selection in response to a previous comment above.

What is the rationale for the chosen sample size? Did the authors perform a priori power calculations?

-Velkey et al (2019) used a total of 78 subjects (males and females with no tests for sex differences) in conditions testing a live-shoal stimulus (vs. either a video stimulus or a mobile-model stimulus). We use this as the rationale for the current N = 82. This information is added to the Subjects sub-section

Plots: please use dot plots instead and show the error bars for both directions (10.1371/journal.pbio.1002128). It is also advisable to use standard deviation instead of standard error of the mean.

-We agree that dot plots can be useful for visualizing variability within certain data sets. However, the article referenced by the reviewer as well as other citing articles indicated this is a better practice for small-n studies; typically, these studies have sample sizes of 4 - 10 subjects. Notably, Weissgerber et al. (2015) state, "The summary statistics shown in bar graphs, line graphs, and box plots are only meaningful when there are enough data to summarize". We argue that bar charts are the best visualization for a meaningful representation of our data, given our sample size of 80-82 subjects. Furthermore, most of our bar charts represent repeated-measures data. Using dot plots with lines to connect each individual subject's points across the levels of the repeated measures would present a confusing mess of several hundred points and line segments on the graph. 

-As the figures are used in the current manuscript to provide the reader with visualizations which support our interpretations of inferential parametric post-hoc comparisons (rather than merely the results of descriptive analyses), we feel that the SEM as a representation of the uncertainty regarding the point-estimation of means is better than the use of the SD which merely indicates the dispersion of the data about the mean in the sample. Therefore, we have decided to retain the use of the SEM, but now show +/- error bars as per Reviewer 2’s request.

Supplementary material cannot be understood on its own. A variable key should be provided. For example, the reader cannot guess whether 1 is the code for males or for females, and the difference between columns Q (InzoneArenaCenterpointCumulativeDurations) and R (InzoneArenaCenterpointCumulativeDuration) is not very obvious.

-We have added a second page to the supplementary data file which includes brief descriptions of each variable along with the value labels for sex and zone parameters.

---

## [Decision Letter · Decision Letter 1]

1 Sep 2022

Female Zebrafish (*Danio rerio*) Demonstrate Stronger Preference for Established Shoals over Newly-Formed Shoals in the Three-Tank Open-Swim Preference Test

PONE-D-22-06149R1

Dear Dr. Velkey,

We’re pleased to inform you that your manuscript has been judged scientifically suitable for publication and will be formally accepted for publication once it meets all outstanding technical requirements.

Kind regards,

Cristiano Bertolucci, Ph.D.

Academic Editor

PLOS ONE

Additional Editor Comments (optional):

Reviewers' comments:

Reviewer's Responses to Questions

**Comments to the Author**

1. If the authors have adequately addressed your comments raised in a previous round of review and you feel that this manuscript is now acceptable for publication, you may indicate that here to bypass the “Comments to the Author” section, enter your conflict of interest statement in the “Confidential to Editor” section, and submit your "Accept" recommendation.

Reviewer #1: All comments have been addressed

Reviewer #2: All comments have been addressed

2. Is the manuscript technically sound, and do the data support the conclusions?

Reviewer #1: Yes

Reviewer #2: Yes

3. Has the statistical analysis been performed appropriately and rigorously? 

Reviewer #1: Yes

Reviewer #2: Yes

4. Have the authors made all data underlying the findings in their manuscript fully available?

Reviewer #1: Yes

Reviewer #2: Yes

5. Is the manuscript presented in an intelligible fashion and written in standard English?

Reviewer #1: Yes

Reviewer #2: Yes

6. Review Comments to the Author

Reviewer #1: The authors have now addressed all the concerns and suggested reccomendations made in the earlier review of the manuscript. the manuscript now reads satisfactorily and has improved in clarity.

Reviewer #2: (No Response)

7. PLOS authors have the option to publish the peer review history of their article (what does this mean?). If published, this will include your full peer review and any attached files.

Reviewer #1: No

Reviewer #2: No

---

## [Editor Report · Acceptance letter]

12 Sep 2022

PONE-D-22-06149R1 

Female Zebrafish (*Danio rerio*) Demonstrate Stronger Preference for Established Shoals over Newly-Formed Shoals in the Three-Tank Open-Swim Preference Test 

Dear Dr. Velkey:

I'm pleased to inform you that your manuscript has been deemed suitable for publication in PLOS ONE. Congratulations! Your manuscript is now with our production department. 

Kind regards, 

on behalf of

Cristiano Bertolucci 

Academic Editor

PLOS ONE